# Comparative Chloroplast Genomes of Six Magnoliaceae Species Provide New Insights into Intergeneric Relationships and Phylogeny

**DOI:** 10.3390/biology11091279

**Published:** 2022-08-28

**Authors:** Lin Yang, Jinhong Tian, Liu Xu, Xueli Zhao, Yuyang Song, Dawei Wang

**Affiliations:** 1Key Laboratory for Forest Resources Conservation and Utilization in the Southwest Mountains of China Ministry of Education, Southwest Forestry University, Kunming 650224, China; 2Key Laboratory for Forest Genetics and Tree Improvement and Propagation in Universities of Yunnan Province, Southwest Forestry University, Kunming 650224, China; 3Department of Forestry, Agricultural College, Xinjiang Shihezi University, Shihezi 832003, China

**Keywords:** Magnoliaceae, chloroplast genome, phylogenomics, intergeneric relationship

## Abstract

**Simple Summary:**

Magnoliaceae is one of the most endangered families of angiosperms. The systematic classification of Magnoliaceae has been controversial for a long time due to minor differences in morphology. In the present study, six new chloroplast genomes of Magnoliaceae were sequenced, and the 37 published chloroplast genomes of the family were subjected to phylogenetic analyses. The results showed that all these chloroplast genomes possess the typical quadripartite structure with a conserved genome arrangement and gene structures, yet their lengths varied due to the expansion/contraction of the IR/SC boundaries. Phylogenetic relationships within Magnoliaceae were determined using complete cp genome sequences. These findings will provide a theoretical basis for adjusting the phylogenetic position of Magnoliaceae at the molecular level.

**Abstract:**

Magnoliaceae plants are industrial tree species with high ornamental and medicinal value. We published six complete chloroplast genomes of Magnoliaceae by using Illumina sequencing. These showed a typical quadripartite structure of angiosperm and were 159,901–160,008 bp in size. A total of 324 microsatellite loci and six variable intergenic regions (Pi > 0.01) were identified in six genomes. Compared with five other genomes, the contraction and expansion of the IR regions were significantly different in *Manglietia grandis*. To gain a more thorough understanding of the intergeneric relationships in Magnoliaceae, we also included 31 published chloroplast genomes of close relative species for phylogenetic analyses. New insights into the intergeneric relationships of Magnoliaceae are provided based on our results and previous morphological, phytochemical and anatomical information. We suggest that the genus *Yulania* should be separated from the genus *Michelia* and its systematic position of should be restored; the genera *Paramichelia* and *Tsoongiodendron* should be merged into the genus *Michelia*; the genera *Pachylarnax* and *Parakmeria* should be combined into one genus. These findings will provide a theoretical basis for adjusting the phylogenetic position of Magnoliaceae at the molecular level.

## 1. Introduction

Chloroplasts are critical plant organelles that play a prominent role in photosynthesis [1]. Chloroplast genomes (cp genomes) are highly conserved because of the genetic replication mechanisms of uniparent inheritance and the relatively high level of genetic variation resulting from the low selective pressure, making them useful for revealing phylogenetic relationships [2]. With the development of Illumina and assembly technologies, the cp genomes of an increasing number of species have been published [3,4,5]. These cp genomes provide valuable information about species identification, trait improvement, genealogical geography and the conservation of endangered species [6,7,8].

Magnoliaceae is one of the most endangered families of angiosperms, and it was listed under Class II National Protection in China [9]. It is considered a key material indispensable for exploring the origin of angiosperms and also an important component of tropical to temperate evergreen broadleaf in deciduous broadleaf forests, which are ecologically important [10]. Magnoliaceae plants are industrial tree species with high medicinal value [11]. The leaves, flowers and bark of them are rich in monoterpenes and sesquiterpenes, which have good anti-tumor-promoting and anti-carcinogenesis activities and are used to treat inflammation and ulceration diseases [12].

The current methods for distinguishing the taxonomic position of Magnoliaceae mainly consider anatomical and morphological aspects [13]. The systematic classification of Magnoliaceae has been controversial for a long time [14]. A total of 12 genera were classified in the narrow concept of Magnoliaceae for the first time by Dandy in 1964 [15], and afterward, it was split into 16 and 18 genera according to the characters of stomatal pores on the leaf epidermis and polygamous flower, respectively [16,17]. A few years later, some taxonomists suggested that Magnoliaceae should be divided into two genera (*Magnolia* L. and *Michelia* L.) based on their main morphological traits, while the remaining 16 genera should be combined with both [18]. In summary, the main controversial differences in the classification of Magnoliaceae are the merging or separation of intergeneric relationships [14]. In our study, we reconstructed the phylogenetic relationship among the genera *Yulania* Spach, *Michelia* L., *Paramichelia* Hu, *Tsoongiodendron* Chun, *Pachylarnax* Dandy and *Parakmeria* Hu & W.C.Cheng by using 37 species of Magnoliaceae to carry out a sequence alignment and phylogenetic analysis of cp genomes. These results provide a molecular-level basis to determine the systematic taxonomic position of Magnoliaceae species.

## 2. Materials and Methods

### 2.1. Plant Materials and DNA Sequencing

The young green and disease-free leaves of 6 species for Magnoliaceae were collected from natural distribution areas (Table 1). The plant species was identified by *Assoc. Prof.* Jianhua Qi (College of Forestry, Southwest Forestry University), and the voucher specimens were stored at the Key Laboratory for Forest Resources Conservation and Utilization in the Southwest Mountains of China Ministry of Education (2020Y18), Southwest Forestry University, Kunming, China. DNA extraction and sequencing were performed according to a previous study by Wang et al. [19].

### 2.2. Chloroplast Genome Assembly and Annotation

The cp genome sequences of *Manglietia dandyi* (MF990567) were used as a reference sequence to assemble the 6 cp genomes of Magnoliaceae using MEGA5.1(Mega Limited, Auckland, New Zealand) [20]. The annotation of the 6 cp genomes was performed via Genious 8.1.3 with sequences of other closely related species. The method of genome annotation was the same as Zheng et al. [21]. The sequences of 6 cp genomes were deposited in GenBank NCBI (MW415418, MW415419, MW415420, MW415421, MW415416 and MW415417). The cp genome map was drawn using OGDRAW37 [22].

### 2.3. Sequence Divergence, Genome Comparison and Single-Sequence Repeat Analysis

The 6 cp genomes of the Magnoliaceae were sequenced performed using the VISualization Tool in Shuffle-LAGAN mode for Alignments [23]. We used the DnaSPv. 5.0 software (J. Rozas et al., Barcelona, The Kingdom of Spain) to set the parameter to a window length size of 600 bp and the distance between each locus to 200 bp to measure nucleotide diversity (Pi) [24]. The 6 cp genome sequences were uploaded to the online IRscope software to visualize their IR/SC boundaries using the .gb format [25]. The simple sequence repeat (SSR) markers were searched by surveying six genomic sequences of the Magnoliaceae using MISAv program (http://genome.lbl.gov/vista/index.shtml, accessed on 15 March 2022) [26].

### 2.4. Phylogenetic Analysis

Sequence alignment was performed using the newly assembled 6 cp genomes and 25 closely related cp genomes, with 6 species of the genera *Illicium* L., *Kadsura* Kaempf. ex Juss. and *Schisandra* Michx. added for analysis, which were downloaded from the NCBI (Appendix A). Phylogenetic analyses were performed according to a study of Wu et al. [27].

## 3. Results

### 3.1. Characteristics of the Six cp Genomes

The six cp genomes of Magnoliaceae are similar to other angiosperms (Table 2 and Figure 1). The complete cp genome is between 159,901 and 160,008 bp in length, exhibiting a classic four-partition structure with an SSC region (18,800–18,803 bp), LSC region (87,753–88,534 bp h) and two IR regions (26,207–26,602 bp). Six cp genomes contained 131 genes (86 protein-coding genes, 37 tRNA genes and 8 rRNA genes), which include 44 photosynthesis genes, 58 translation-related genes and 11 other genes (Appendix A).

### 3.2. Comparative Genomic, IR Expansion and Contraction, and SSR Analysis

To investigate the levels of sequence polymorphism, the six cp genomes of Magnoliaceae species were compared (Figure 2). The results showed that the structures, orders and contents of these six cp genomes were all conserved. The Pi values of these six genomes ranged from 0 to 0.0153. Although aligned sequences showed relatively low divergences, some hotspot regions with high variation were also identified. The variable regions with Pi exceeding 0.01 in the six cp genomes were *ndhF-trnL-UAG*, *ndhD-ndhE*, *rpl32-trnL-UAG*, *petG-psaJ*, *psaC-ndhA*, *trnF-* and *GAA-ndhK* (Figure 3).

The six CP genomes’ IR/LSC and IR/SSC boundary structures were compared (Figure 4). The results showed that the IR boundaries of the cp genomes of the six Magnoliaceae species were comparatively conserved. Only the *rpl2* gene of *Manglietia*
*grandis* expanded to the LSC region, with an expansion length of 308 bp, and the *rpl2* genes of the remaining five species were in the IRb region. Among them, the distributions of genes on the IRb/SSC and SSC/IRa boundaries were similar for the *ndhF* and *ycf1* genes, and the length of the *ycf1* gene on the SSC/IRa boundary ranged from 5558 to 5594 bp, all of which were pseudogenes. The characteristics of SSRs in six cp genomes were analyzed, a total of 324 repeats were certified in six genomes, and most SSRs included the A/T rather than the G/C motif (Figure 5b and Appendix A). Mononucleotide repeats were the most abundant SSR in all the species; pentanucleotide repeats were the least abundant. The analysis of long repeats in six species revealed more forward and palindromic repeats than reverse and complementary repeats (Figure 5C,D). 

### 3.3. Phylogenetic Analysis

Phylogenetic analysis of six species was performed using the ML method (Figure 6); the results showed that most of the nodes had 100% bootstrap values. The phylogenetic tree showed that the 37 species of Magnoliaceae can be broadly divided into two clusters. Among them, the genera *Yulania*, *Paramichelia*, *Michelia*, *Tsoongiodendron*, *Alcimandra* Dandy, *Pachylarnax*, *Parakmeria*, *Woonyoungia* Y.W.Law, *Manglietia* Blume, *Talauma* Juss. and *Liriodendron* L. were clustered into one group, and the genera *Illicium*, *Kadsura* and *Schisandra* were also clustered into one group. In our phylogenetic tree, *Pachylarnax, Parakmeria* and *Michelia* were closely related to *Paramichelia* and *Tsoongiodendron*, but the genera *Illicium*, *Kadsura* and *Schisandra* are not clustered into a group with Magnoliaceae species. In addition, we also compared the Flora Reipubicae Popularis Sinicae (FRPS) and Flora of China (FOC) plant classifications in Magnoliaceae, finding that the taxonomic statuses of the genera *Paramichelia*, *Tsoongiodendron*, *Pachylarnax* and *Parakmeria* were different.

## 4. Discussion

Here, for the first time, we present cp genomes for six Magnoliaceae species, including four *Manglietia* species and two *Yulania* species. These cp genomes are consistent with the characteristics of most angiosperm species [28], and did not differ significantly from each other in terms of structure and length (159,901–160,008 bp). In addition, we found that the mean contents of AT and GC in these six cp genomes were 61.7% and 39.3%. In the genome, the higher the AT content, the lower the DNA density, and the sequences were more prone to denaturation and mutation [29]. Therefore, we speculated that the six cp sequences of Magnoliaceae were somewhat mutagenic and their chloroplast gene sequences might be more prone to variation than those of other species.

The results of the IR boundary analysis showed that the contraction of the IR region (26,207 bp) was the most pronounced in *Manglietia grandis*, with an expansion of the *rpl2* gene in its IR to LSC region of 308 bp, while the *rpl2* genes of the other five species were intact and located in the IR region. This indicated that the boundary change of LSC /IR is the dominant factor affecting the expansion and contraction of the cp genome IR region of *Manglietia grandis*. However, such an expansion is small, and no important expansions or contractions were observed in these cp sequences. This result is similar to the expansion of the chloroplast genomes of other Magnoliaceae species in the IR region [30], but different from the contraction of Zingiberaceae and Arecaceae [31,32]. This indicates that different species have evolved under the influence of different factors, resulting in different degrees of expansion and contraction of IR/SC boundaries, thus showing the diversity in genome length and boundaries [33].

The varied SSRs in cp genomes have a greater taxonomic distance between them than nuclear and mitochondrial genomes; they are widely used in studies of the genetic diversity and germplasm resources of plant populations [34]. We identified 324 SSRs in cp genomes of six Magnoliaceae species, most of which had mononucleotide repeats composed of A/T. These SSRs can be used to develop microsatellite markers for genetic diversity and evolution analyses [35]. We also screened a total of seven highly variable regions through nucleotide diversity analysis. Among them, four were located in the LSC region and three in the SSC region. This indicates that the LSC and SSC regions of these six Magnoliaceae species have high nucleotide variability, and these highly variable regions can be used as potential polymorphic molecular markers for evolutionary studies [36].

These six cp genome sequences were phylogenetically analyzed with their 31 rela-tives; the results showed that species of Magnoliaceae clustered in a group, and the genera *Illicium*, *Schisandra* and *Kadsura*, which do not belong to Magnoliaceae, were divided into a separate group. This result is consistent with the classification of Angiosperm Phylogeny Group (APG IV) system [37]. Meanwhile, the most of nodes had high bootstrap values in our phylogenetic tree, and the results of phylogenetic analysis for monophyletic genera are consistent with previous studies, indicating that the phylogenetic tree in this study is reliable [13,38]. The aim of this study was to determine the intergeneric relationships within Magnolioideae, as the systematic classification of Magnoliaceae has been controversial for a long time [14]. It has previously been demonstrated that the genus *Yulania* is included in the genus *Michelia* due to its pre-branching characteristics [39]. However, the contents of volatile oils obtained from flower and the pit vessel characteristics of wood of these two genera were significantly different in subsequent studies [40]. In particular, reproductive isolation was discovered due to the discontinuity of geographical distribution of the two genera; the genus *Yulania* was separated from the genus *Michelia* [41,42]. This result is consistent with the phylogenetic analysis conducted in our study. Furthermore, it was consistent with previous conclusions inferred from *Matk* and *ndhF* sequences [43]. Similar results were reported for the Bupleurum family, with new insights into its phylogenetic status provided through assessing the cp genomes and morphological characteristics of fruits and leaves [34,44]. We thus suggest that the genus *Yulania* should be separated from the genus *Michelia*, and its systematic position should be restored.

In the present study, the genera *Paramichelia*, *Tsoongiodendron* and *Michelia* were clustered into one clade, which is identical to the results of another phylogenetic analysis based on molecular markers [45]. This strongly supports the idea of a close relationship between these three genera. It has been argued that the genera *Paramichelia* and *Tsoongiodendron* should be separated from the genus *Michelia* according to the different characters of the ripe fruit carpels [17]. This tiny difference is considered by traditional taxonomists to be the result of parallel evolution [46]. In other words, these three genera come from the same ancestor and therefore show the same trend in evolution [47]. Based on all this evidence, we share the view that the genera *Paramichelia* and *Tsoongiodendron* should be merged into the genus *Michelia*. Similarly, Flora of China suggested adjusting the genera *Paramichelia* and *Tsoongiodendron* to genus-level status in the systematic position [48].

The genus *Pachylarnax* was established based on its polygamous flower [49], and it is considered to be more closely related to the genus *Manglietia* [50]. This argument was not consistent with the result of the phylogenetic analysis in our study; we suggested that, compared with *Manglietia,* the genus *Parakmeria* is more closely related to *Pachylarnax*. Meanwhile, this view is also consistent with the results of the phylogenetic analysis using the B-class MADS-box gene [51]. Additionally, the genera *Pachylarnax* and *Parakmeria* both have the high-taxonomic-value characteristic of curling young leaves [52]. We thus recommend that the genera *Pachylarnax* and *Parakmeria* should be combined into one genus. Furthermore, based on all the results related to phylogenetic relationships, we compared the two classifications and found that the FRPS can locate the species attribution more precisely than FOC in Magnoliaceae.

## 5. Conclusions

This study reports the complete cp genome sequence of six Magnoliaceae species: *M. crassipes*, *M. grandis*, *M. hookeri*, *M. ventii, Y. praecocissima* and *Y. soulangeana*. New insights into the intergeneric relationships in the Magnoliidae family are provided by combining our findings with previous studies. We recommend that the genus *Yulania* should be separated from the genus *Michelia*, and the systematic position of *Yulania* should be restored; the genera *Paramichelia* and *Tsoongiodendron* should be merged into the genera *Michelia*; and the genera *Pachylarnax* and *Parakmeria* should be combined into one genus. These results provide a theoretical foundation for the phylogenetic position of Magnoliaceae at the molecular level.

## Figures and Tables

**Figure 1 biology-11-01279-f001:**
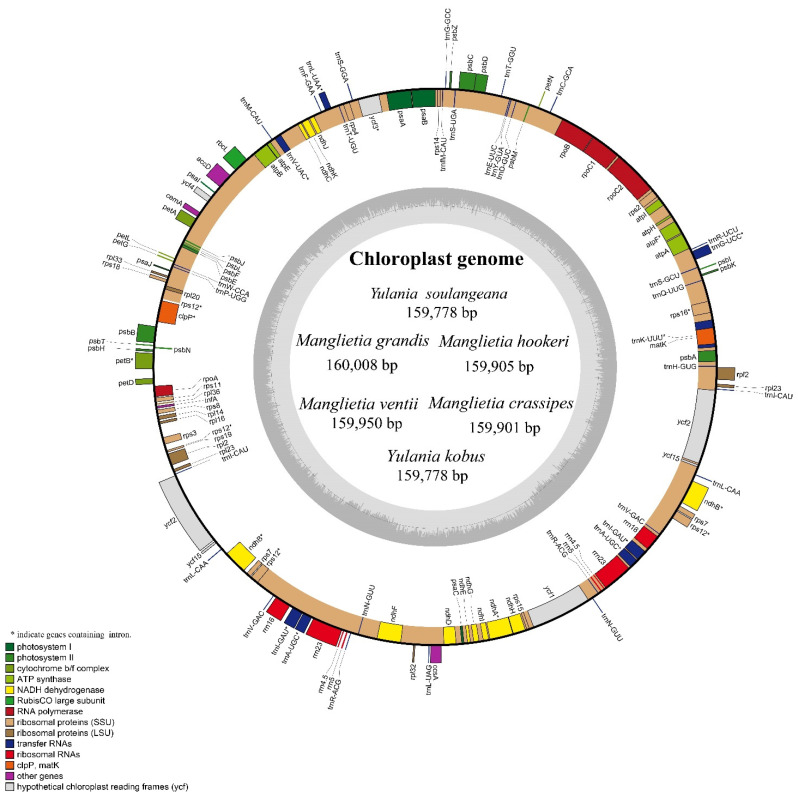
Structural map of the six chloroplast genomes of Magnoliaceae species.

**Figure 2 biology-11-01279-f002:**
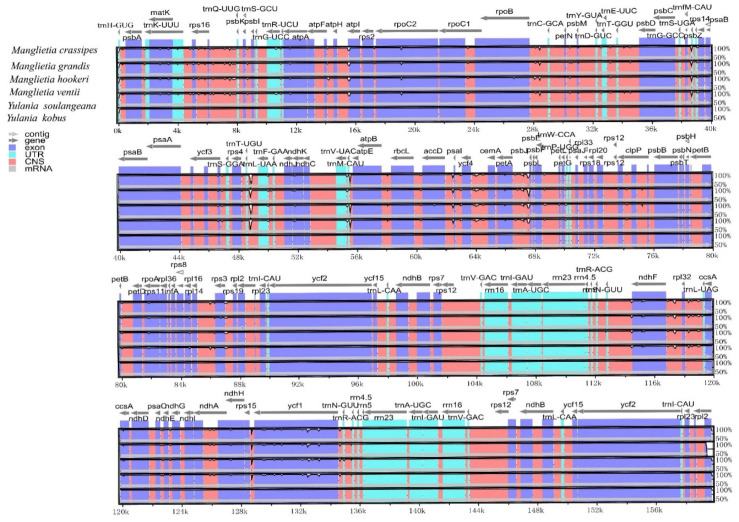
Alignment of whole chloroplast genome sequences from the six Magnoliaceae species.

**Figure 3 biology-11-01279-f003:**
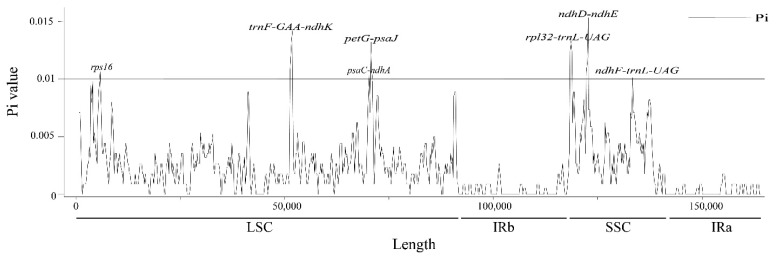
Sliding window analysis of the whole chloroplast genome nucleotide diversity (Pi) of the six Magnoliaceae species.

**Figure 4 biology-11-01279-f004:**
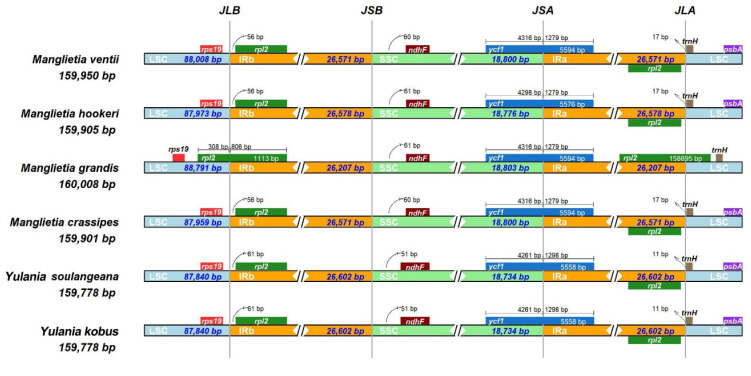
Comparison of the border regions of the six chloroplast genomes of Magnoliaceae. Note: Different genes are denoted by colored boxes. The gaps between the genes and the boundaries are indicated by the base lengths (bp).

**Figure 5 biology-11-01279-f005:**
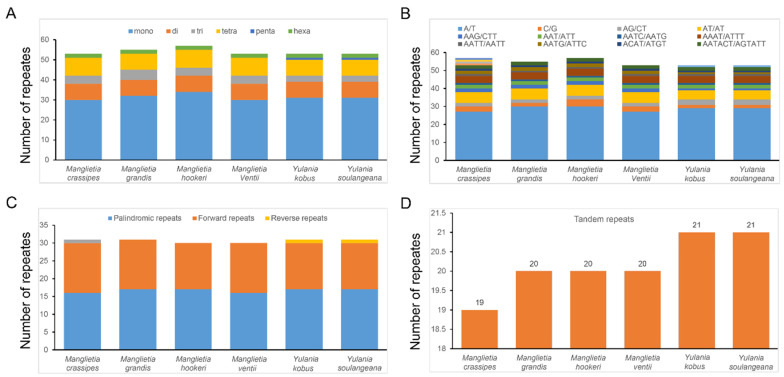
Comparison of repeats in six species of Magnoliaceae family. Note: (**A**) The type frequency of different SSR types. (**B**) The type frequency of SSR motifs in different repeat class types. (**C**) The type frequency of different repeat types. (**D**) The type frequency of dispersed repeat sequences.

**Figure 6 biology-11-01279-f006:**
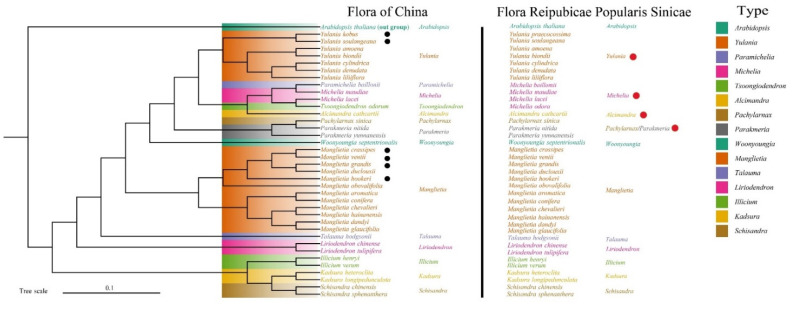
Maximum likelihood (ML) phylogenetic tree based on 37 complete chloroplast genomes of Magnoliaceae. Note: Black circles indicate the six species of Magnoliaceae in this study. The red circles indicate the genera whose phylogenetic positions were discussed in this study.

**Table 1 biology-11-01279-t001:** The sampling area and information of six species of the Magnoliaceae family.

Genus	Species	Protection Grade	Sampling Area	Longitude/Latitude
*Manglietia*	*Manglietia crassipes* Y.L.Law	-	Guangxi, China	109°50′ E/23°40′ N
	*Manglietia grandis* Hu & W.C.Cheng	II	Yunnan, China	104°33′ E/22°48′ N
	*Manglietia hookeri* Cubitt & W.W.Sm.	-	Yunnan, China	99°55′ E/21°10′ N
	*Manglietia ventii* N.V.Tiep.	II	Yunnan, China	102°10′ E/24°23′ N
*Yulania*	*Yulania kobus* (DC.) Spach	-	Yunnan, China	102°10′ E/24°23′ N
	*Yulania soulangeana* (Soul.-Bod.) D.L.Fu	-	Yunnan, China	102°10′ E/24°23′ N

Notes: II: Endangered

**Table 2 biology-11-01279-t002:** Summary of chloroplast genome characteristics of six Magnoliaceae chloroplast genomes.

Species	*Manglietia crassipes*	*Manglietia grandis*	*Manglietia hookeri*	*Manglietia ventii*	*Yulania kobus*	*Yulania* *soulangeana*
Total length (bp)	159,901	160,008	159,905	159,950	159,778	159,778
LSC length (bp)	87,959	88,534	87,973	88,008	87,840	87,753
SSC length (bp)	18,800	18,803	18,776	18,800	18,734	18,734
IR length (bp)	26,571	26,207	26,578	26,571	26,602	26,602
Overall GC content (%)	39.3	39.3	39.3	39.3	39.3	39.3
Total gene number	131	131	131	131	131	131
GenBank accession	MW415418	MW415419	MW415420	MW415421	MW415416	MW415417

## Data Availability

The datasets generated during and/or analyzed during the current study are available in the NCBI repository, [https://www.ncbi.nlm.nih.gov/] (accessed on 5 March 2022). The datasets generated during and/or analyzed during the current study are available from the corresponding author on reasonable request. All data generated or analyzed during this study are included in this published article [and its Appendix A].

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
