# Peer review of "Comparative Chloroplast Genomes of Six Magnoliaceae Species Provide New Insights into Intergeneric Relationships and Phylogeny"

_biology, 2022, doi:10.3390/biology11091279_

Round 1
Reviewer 1 Report
1. Some recent key references should be introduced, compared and discussed with the current study. They are:
Wang, Y.-B., Liu, B.-B., Nie, Z.-L., Chen, H.-F., Chen, F.-J., Figlar, R.B. and Wen, J. (2020), Major clades and a revised classification of Magnolia and Magnoliaceae based on whole plastid genome sequences via genome skimming. J. Syst. Evol., 58: 673-695. https://doi.org/10.1111/jse.12588
Dong, S.-S., Wang, Y.-L., Xia, N.-H., Liu, Y., Liu, M., Lian, L., Li, N., Li, L.-F., Lang, X.-A., Gong, Y.-Q., Chen, L., Wu, E. and Zhang, S.-Z. (2022), Plastid and nuclear phylogenomic incongruences and biogeographic implications of Magnolia s.l. (Magnoliaceae). J. Syst. Evol., 60: 1-15. https://doi.org/10.1111/jse.12727
Guzmán-Díaz S, Núñez FAA, Veltjen E, Asselman P, Larridon I, Samain M-S. Comparison of Magnoliaceae Plastomes: Adding Neotropical Magnolia to the Discussion. Plants. 2022; 11(3):448. https://doi.org/10.3390/plants11030448
2. "choloplast genome" should be replaced by "plastome" in the whole text because chloroplast is only one type of plastid.
3. Plant materials should be given with voucher specimens and deposited places (such as a herbarium). If voucher specimens were not collected, they should be recollected.
4. In Table 1, the protection grade should be updated according to the List of National Key Protected Wild Plants in China that was issued in 2021. See the link: http://www.gov.cn/zhengce/zhengceku/2021-09/09/content_5636409.htm
5. Species included for phylogenetic analysis should be carefully considered. There are currently more than 100 choloplast genomes for Magnoliaceae. One choice is to include as many as possible. Another choice is to select representative species for each clade based on previous studies, but to include as many species for those clades that were concerned in this study.
6. Method for phylogenetic analysis should be given with more details and should be improved. The current analysis just using MEGA is considered too simple. Other softwares such as IQTREE, RaxML are suggested. The tree inference method (ML, MP or BI) and bootstrap analyses should also be given with more details.
Reviewer 2 Report
Review comments
The authors obtained the chloroplast genomes of 6 Magnoliaceae species and conducted phylogenetic analysis. The results showed the 6 species have structural variations of the chloroplast genome, but its difference is small. The resultant phylogenetic tree implies that several genera such as Tsoongiodendron, Pachylarnax, and Parakmeria need taxonomic reconsideration, and genus Yulania should be separated from genus Michelia. This manuscript is acceptable for this journal. But I propose several minor comments.
・The tense of words is sometimes wrong and English expression is awkward throughout the manuscript, so I recommend the authors to use an English editing service.
For example, was→is (L.51, 52), was→has been (L.52), were→are (L.56), clear→clarify (L.208) and so on.
・Phylogenetic position → taxonomic position (L. 59)
・Genus Michela includes 70 species, and Parakmeria includes 5 species, but this study treated only two species, respectively. So, I think the authors should refer to this point and should be cautious.
